# Comparison between EUCAST Broth Microdilution and MIC Strip Test in Defining Isavuconazole In Vitro Susceptibility against *Candida* and Rare Yeast Clinical Isolates

**DOI:** 10.3390/antibiotics12020251

**Published:** 2023-01-26

**Authors:** Maddalena Calvo, Guido Scalia, Concetta Ilenia Palermo, Salvatore Oliveri, Laura Trovato

**Affiliations:** 1U.O.C. Laboratory Analysis Unit, A.O.U. “Policlinico-San Marco”, 95123 Catania, Italy; 2Department of Biomedical and Biotechnological Sciences, University of Catania, 95123 Catania, Italy

**Keywords:** isavuconazole, yeasts, EUCAST, antifungal susceptibility

## Abstract

Isavuconazole is a new broad-spectrum triazole, with significant in vitro activity against yeasts. Isavuconazole in vitro susceptibility can be evaluated through broth microdilution as a reference method. Considering difficulties in equipping such methods in a laboratory routine, a commercial MIC Strip test has been designed. This study aims to implement data about isavuconazole in vitro activity and compare EUCAST broth microdilution and MIC Strip test in defining yeast isavuconazole susceptibility. The study involved 629 isolates from positive blood cultures (January 2017–December 2021). The identified species were *C. albicans* (283), *C. glabrata* (53), *C. krusei* (23), *C. tropicalis* (68), *C. parapsilosis* complex (151), *C. guilliermondii* (12), *C. famata* (6), *S. cerevisiae* (12), *C. neoformans* (5), *S. capitata* (12), and *Rhodotorula* species (4). All the isolates were tested with EUCAST microdilution and MIC Strip methods. The total essential agreement between these two methods was 99.3%. As a result, we can consider that both methods are useful in testing isavuconazole susceptibility. Proposed cut-off values (P-ECOFF) were calculated using ECOFFinder software. Further studies could lead to either definitive E-COFF or clinical breakpoints, which represent the most important categorization tool of the laboratory data, allowing a better insertion of an antimicrobial drug in clinical practice.

## 1. Introduction

Isavuconazole was released in 2015 and it is currently known as the most recent triazole compound, whose activity is against the 14-α-demethylase, a membrane protein involved in ergosterol biosynthesis. Isavuconazole structure is like voriconazole one, but it differs for an added sidearm which expands its spectrum [1,2,3,4]. Isavuconazole can be considered a new broad-spectrum triazole, with significant in vitro activity against yeasts, moulds, and dimorphic fungi. Furthermore, several studies confirm isavuconazole clinical efficacy in case of invasive aspergillosis, mucormycosis and candidaemia [5,6,7]. Candidaemia and invasive candidiasis have the highest morbidity and mortality rates among invasive fungal infections, therefore *Candida* spp. isolates hold the primacy among aetiological agents. Despite this, some rare yeasts (*Saprochaete capitata*, *Saccharomyces cerevisiae*, *Cryptococcus neoformans*, *Rhodotorula* species) have emerged as a cause of invasive fungal infection over the past decade [8,9]. Due to increased episodes of fungemia, the use of antifungal drugs has also been increased. Consequently, due both to the high rate of invasive fungal infections and the possible finding of drug resistance in relevant clinical isolates, studies on antifungal drugs continue to be encouraged. Isavuconazole in vitro susceptibility can be evaluated through broth microdilution following EUCAST (European Committee on Antimicrobial Susceptibility Testing) or CLSI (Clinical and Laboratory Standards Institute) principles, with some differences in procedure and results’ interpretation [10,11,12]. Although broth microdilution is precisely defined, it fronts possible difficulties in equipping into a laboratory diagnostic routine. For this reason, a commercial method for immediate use has been designed: the Gradient test principle can be used for the application of the MIC Strip test [13,14,15]. According to the latest EUCAST and CLSI guidelines, neither epidemiological cut-off nor clinical breakpoints regarding yeasts and isavuconazole are still defined [10]. This study aims to implement literary data about isavuconazole in vitro activity [16,17,18], but also to compare a reference method (EUCAST broth microdilution) and a commercial method (MIC Strip test) in defining isavuconazole in vitro susceptibility regarding yeasts, as previous studies tried to do [19]. If the commercial method shows a good agreement with the reference one, then it could also become a useful technique easily practicable in a laboratory routine.

## 2. Results

### 2.1. Isolates Distribution

A varied distribution of fungemia episodes was observed considering different hospitalization wards (Figure 1). Most fungemia cases were from the Internal Medicine clinic (48%), followed by the Intensive Care Unit (23%) and Surgery wards (16%). The Hematology ward hosted 7% of fungemia episodes, while the lowest percentages were from pneumology and Infectious Diseases wards (4% and 2% respectively). Some changes had been highlighted relatively to different yeast species distribution in the involved hospital wards. *C. albicans*, *C. tropicalis*, *C. parapsilosis* complex and *S. cerevisiae* strains were most founded in the Internal Medicine clinic; *S. capitata* strains were isolated from hematologic patients, while *C. neoformans* strains came predominantly from the Infectious Diseases ward.

### 2.2. EUCAST Broth Microdilution

After the EUCAST broth microdilution application, statistical and microbiological calculations were performed (Table 1). For *C. albicans* strains, a geometric mean of 0.010 and a mode of 0.008 mg/L were reported. MIC_50_ and MIC_90_ values were respectively 0.008 and 0.016 mg/L. A total MIC range of 0.008–1 mg/L was observed.

For *C. glabrata* strains, a geometric mean of 1.898 and a mode of 2 mg/L were observed. MIC50 and MIC90 had the same value (2 mg/L). Every MIC value has returned within 1–2 mg/L. For *C. krusei* isolates, a geometric mean of 0.613 and a mode of 0.5 mg/L were reported. MIC50 and MIC90 had the same value (0.5 mg/L). A total MIC range of 0.5–2 mg/L was observed.

*C. tropicalis* strains showed a geometric mean of 0.010 and a mode of 0.008 mg/L. MIC_50_ and MIC_90_ were equal to 0.008 mg/L. A MIC range of 0.008–2 mg/L was observed. *C. parapsilosis* complex isolates showed a geometric mean of 0.014 and a mode of 0.016 mg/L. MIC_50_ had the value of 0.016 mg/L, while MIC_90_ had the value of 0.032 mg/L. A MIC range of 0.008–0.125 mg/L was observed. The MICs for *C.. krusei* ATCC 6258 and *C. parapsilosis* ATCC 22019 were in the recommended ranges.

*C. guilliermondii* strains showed a geometric mean of 0.009 mg/L and a mode of 0.008 mg/L. MIC_50_ and MIC_90_ had the value of 0.008 mg/L. A MIC range of 0.008–0.016 mg/L was detected. *C. famata* strains showed a geometric mean and a mode of 0.006 mg/L. MIC_50_ and MIC_90_ had a value of 0.006 mg/L. All detected MIC values were equal to 0.006 mg/L.

For *S. cerevisiae* strains, a geometric mean, and a mode of 0.03 mg/L were calculated. MIC_50_ and MIC_90_ values were equal to 0.03 mg/L. All detected MIC values were equal to 0.03 mg/L. *C. neoformans* isolates showed a geometric mean and a mode of 0.016 mg/L. MIC_50_ and MIC_90_ had the same value of 0.016 mg/L. All detected MIC values were equal to 0.008 mg/L.

For *S. capitata* strains, a geometric mean of 1.887 and a mode of 2 mg/L. MIC_50_ and MIC_90_ had the same value (2 mg/L). Every MIC value has returned within 2 mg/L. *Rhodotorula* spp. strains showed a geometric mean and a mode of 0.016 mg/L. MIC_50_ and MIC_90_ were both equal to 0.016 mg/L. Every detected MIC value was equal to 0.016 mg/L.

### 2.3. MIC Strip Test

After the MIC Strip test application, statistical and microbiological calculations were performed (Table 1). For *C. albicans* strains, a geometric mean of 0.013 and a mode of 0.008 mg/L were reported. MIC_50_ and MIC_90_ values were respectively 0.004 and 0.032 mg/L. A total MIC range of 0.004–0.125 mg/L was observed.

For *C. glabrata* strains, a geometric mean of 1.877 and a mode of 2 mg/L were observed. MIC_50_ and MIC_90_ had a value of 2 mg/L. Every MIC value has returned within 0.5–4 mg/L. For *C. krusei* isolates, a geometric mean of 0.525 and a mode of 0.5 mg/L were reported. MIC_50_ and MIC_90_ had the same value (0.5 mg/L). A total MIC range of 0.5–1 mg/L was observed.

*C. tropicalis* strains showed a geometric mean of 0.006 and a mode of 0.004 mg/L. MIC_50_ and MIC_90_ were equal to 0.004 mg/L. A MIC range of 0.004–0.38 mg/L was observed. *C. parapsilosis* complex isolates showed a geometric mean of 0.025 and a mode of 0.032 mg/L. MIC_50_ had a value of 0.016 mg/L, while MIC_90_ had a value of 0.032 mg/L. A MIC range of 0.004–0.032 mg/L was observed.

*C. guilliermondii* strains showed a geometric mean of 0.019 mg/L and a mode of 0.016 mg/L. MIC_50_ and MIC_90_ had the value of 0.016 mg/L. A MIC range of 0.016–0.032 mg/L was detected. *C. famata* strains showed a geometric mean and a mode of 0.008 mg/L. MIC_50_ and MIC_90_ had a value of 0.008 mg/L. All detected MIC values were equal to 0.008 mg/L.

For *S. cerevisiae* strains, a geometric mean, and a mode of 0.004 mg/L were calculated. MIC_50_ and MIC_90_ values were equal to 0.004 mg/L. All detected MIC values were equal to 0.004 mg/L. *C. neoformans* isolates showed a geometric mean and a mode of 0.008 mg/L. MIC_50_ and MIC_90_ had the same value of 0.008 mg/L. All detected MIC values were equal to 0.008 mg/L.

For *S. capitata* strains, a geometric mean, and a mode of 1 mg/L were reported. MIC_50_ and MIC_90_ had the same value (1 mg/L). All detected MIC values were equal to 1 mg/L. *Rhodotorula* strains showed a geometric mean and a mode of 0.008 mg/L. MIC_50_ and MIC_90_ were both equal to 0.008 mg/L. Every detected MIC value was equal to 0.008 mg/L.

The MICs for *C. krusei* ATCC 6258 and *C. parapsilosis* ATCC 22019 were in the recommended ranges. Some literature data reported that MIC Strip test for azoles could lead to difficulties: *C. albicans*, *C. glabrata* and *C. tropicalis* might generate microcolonies within the inhibition halo [20]. In the case of microcolonies found during the study, MIC was evaluated as the concentration at which a growth reduction of at least 80% could be reported.

### 2.4. Statistical Analysis

The essential agreement calculation is summarized in Table 2. The total essential agreement percentage was 99.3% (Figure 2), while some differences had emerged relatively to distinction by species. Among *Candida* species, *C. albicans* showed an essential agreement of 99.2%, while *C. glabrata* one was 98.1%. *C. krusei* and *C. parapsilosis* complex essential agreement was 100%. Also *C. famata*, *C. guilliermondii* and *C. incospicua* showed an essential agreement equal to 100%. The lowest value was detected for *C. tropicalis* (95.6%). For rare yeast species such as *S. capitata*, *Rhodotorula* species, *C. neoformans* and *S. cerevisiae* essential agreement values were equal to 100%. A categorical agreement was not possible to establish because of the absence of clinical breakpoints. Consequently, major errors (MEs) or very major errors (VMEs) have not been calculated.

### 2.5. P-ECOFF Calculation

To deepen the importance of in vitro susceptibility studies, a proposed cut-off value (P-ECOFF) was calculated using ECOFFinder software, based on the 99.9% subset of MICs. For *C. albicans*, a P-ECOFF of 0.125 mg/L was determined. The percentage of isolates presenting a MIC higher than P-ECOFF were 0.70% and 0% considering respectively EUCAST and MIC Strip method. A P-ECOFF of 16 mg/L was calculated for *C. parapsilosis* complex and *C. krusei*. No isolates presented a MIC higher than P-ECOFF for these species. For *C. guilliermondii*, a P-ECOOF equal to 16 mg/L was determined. No isolates presented a MIC higher than that value for this species. A P-ECOFF of 16 mg/L was established for *C. tropicalis*. No isolates presented a MIC higher than that value for this species. For *C. glabrata*, *C. famata*, *S. capitata*, *S. cerevisiae*, *C. neoformans* and *Rhodotorula* species was not possible to establish a P-ECOFF, due to the limited number of MIC distributions detected during our in vitro susceptibility tests.

## 3. Discussion

The study aimed to enrich a few literary data available about isavuconazole in vitro activity against yeasts, comparing EUCAST broth microdilution and MIC Strip test in defining yeasts’ isavuconazole in vitro susceptibility. Azoles, drugs that have long been used in the treatment of invasive fungal infections, might face resistance episodes, or a limited spectrum of action. Isavuconazole, a new generation azole, shows an interesting pharmacological profile and a broad spectrum which allows its use in several therapeutic schemes. For instance, isavuconazole received FDA approval for the treatment of invasive aspergillosis and mucormycosis [8]. Moreover, the latest guidelines for the management of candidiasis, released by the Infectious Diseases Society of America (IDSA), highlight the isavuconazole non-inferiority in comparison with echinocandins [20]. Although this premise, isavuconazole has not been approved yet for candidiasis and candidaemia treatment. The importance to produce in vitro susceptibility studies is supported by the increasing resistance rates among fungal isolates. *Candida* species show intrinsic resistance episodes such as fluconazole resistance in *C. krusei* and reduced fluconazole susceptibility in *C. glabrata*. In addition, *C. tropicalis*, *C. albicans* and *C. parapsilosis* may acquire azole resistance after a prolonged drug exposition [21]. Echinocandin resistance is also increasing, especially in *C. glabrata* and *C. albicans*. This kind of resistance is caused by prolonged echinocandin administration in the case of fungemia [22]. Furthermore, rare yeast isolates could also express antimicrobic resistance. First, *S. capitata* is intrinsically resistant to echinocandins and highly resistant to fluconazole [23]. Furthermore, *C. neoformans* is intrinsically resistant to echinocandins due to its cell membrane structure [24]. Both Candida species and rare yeasts can cause invasive fungal infections, whose treatment could be complicated by possible resistance episodes. On these premises, it is essential to promote studies about newer antifungal drugs, such as isavuconazole, and their potential against yeast isolates.

According to other documented and published data [25,26,27], our study’s results confirm a significant isavuconazole in vitro activity against *C. albicans*, *C. tropicalis*, *C. parapsilosis* complex, *C. guilliermondii*, and *C. famata*. Otherwise, higher MICs were reported for *C. krusei* and *C. glabrata*. Among rare yeast isolates, *S. cerevisiae*, *C. neoformans* and *Rhodotorula* species revealed a high rate of susceptibility to isavuconazole, as well as *S. capitata* strains. The essential agreement between broth microdilution and MIC Strip test was high, so we can consider that both methods are useful in testing isavuconazole susceptibility. However, the essential agreement percentage slightly changed when reasoning in a species-specific sense. Specifically, very high values were reported for all the species, including *C. tropicalis* isolates which showed the lowest value (95.6%). This percentage has been also considered optimal.

Overall, all the percentages appear to be significant, and this correspondence suggests the possibility to test isavuconazole in vitro susceptibility with the MIC Strip test, due to a high agreement with the EUCAST reference method. However, further studies should be performed to encourage this suggestion, especially regarding rare yeasts. For example, *S. cerevisiae* MIC results from broth microdilution are concordant with what is reported in the literature, but we don’t have literary data enough regarding MIC Strip test with which to compare those obtained by us for this species [28,29]. Consequently, we need a major number of isolates to establish the effectiveness of the commercial method, considering its distance from the reference method during our analyses. Furthermore, *S. capitata* is a yeast with very singular growth times and characteristics, and we could attribute to these peculiarities some possible discrepancies between the commercial method and the reference method. However, MIC values reported from our EUCAST broth microdilution were concordant with the literature data [30].

Finally, we need to remember that a few considerations can be said about *C. famata* and *C. guilliermondii*: further studies will be necessary to establish if a MIC Strip test could be inserted in the laboratory routine due to a high agreement with reference methods. Our results about these species were certainly positive but also based only on a little selection of clinical isolates. Additional investigations will be encouraged to compare isavuconazole MIC results to other azoles MIC for all the included species. These species could be processed through EUCAST broth microdilution for azoles which are consolidated in the clinical practice such as fluconazole, analyzing differences in MIC ranges and median values. Our data about E-COFF were just preliminary because E-COFF establishment requires huge studies with a major number of isolates and research centres. Further studies could lead to either definitive E-COFF or clinical breakpoints, which allow categorization of the laboratory data. Clinical breakpoints could favour a greater inclusion of isavuconazole in clinical practice, considering that active trials demonstrated its non-inferiority relative to echinocandins for the primary treatment of fungemia and invasive fungal infections [31]. Clinical trials have also been established to test isavuconazole safety against invasive fungal infections among paediatric patients [32]. Moreover, isavuconazole has been compared to voriconazole in invasive aspergillosis, showing high effectiveness [33]. In conclusion, isavuconazole could represent a promising therapeutic option in most invasive fungal infection cases. This conclusion encourages the trend to produce further susceptibility studies about this drug.

## 4. Materials and Methods

### 4.1. Sample Size

The study involved 629 yeast isolates from positive blood cultures of hospitalized patients at the University Hospital in Catania. The period in which the selected patients were hospitalized fluctuated between January 2017 and December 2021. All the isolates were taken from a mycoteca, inoculated in Sabouraud Dextrose agar plates, and added with 2% of glucose (Vakutest Kima, Arzergrande, Italy). Plates were incubated for at least 24–72 h at a temperature of 37 °C. Species or genus identification were confirmed through MALDI Biotyper^®^ Sirius System (Bruker, Billerica, MA, USA) for all the selected isolates. The following microorganisms were identified: *C. albicans* (283), *C. glabrata* (53), *C. krusei* (23), *C. tropicalis* (68), *C. parapsilosis* complex (151), *S. cerevisiae* (12), *C. neoformans* (5), *S. capitata* (12), *Rhodotorula* species (4), *C. famata* (6), and *C. guilliermondii* (12).

### 4.2. EUCAST Broth Microdilution

For isavuconazole in vitro susceptibility through the microdilution method, EUCAST criteria were applied [13]. An RPMI 1640 medium (SSI Diagnostica, Hillerød, Denmark) added with 2% of glucose and buffered with 3-N-morpholino-propane sulfonic acid (MOPS) was prepared. Colonies from identified microorganisms were dissolved into 3 mL of sterile distilled water and mechanically homogenized at the speed of 2000 rpm for 15 s. A 0.5 MacFarland standard was obtained. Isavuconazole powder (Basilea Pharmaceutica Ltd., Basel, Switzerland), was dissolved in dimethyl sulfoxide (DMSO) to reach a concentration of 1600 mg/L, ideal to keep the antifungal range within 0.008–4 mg/L. ATCC control strains such as *C. krusei* (6528) and *C. parapsilosis* (22,019) were also tested. Growth control and sterility control were added for each tested microorganism. Microplates were incubated at 37 °C for 24 h, except for *C. guillermondii* and *C. parapsilosis* complex tests, whose incubation was prolonged up to 48 h. A spectrophotometric reading at the wavelength of 530 nm was performed to obtain MIC results. The lowest isavuconazole concentration in which a 50% growth reduction (compared to the growth control) was reported, was identified as the isavuconazole MIC.

### 4.3. MIC Strip Method

The manufacturer’s instructions were accurately followed for isavuconazole in vitro susceptibility through the MIC Strip method [14]. Nitrocellulose strips soaked with increasing concentrations (0.002–32 mg/L) of the selected antifungal (MTS, Liofilchem s.r.l., Roseto degli Abruzzi, Italy) were placed into an RPMI 1640 MOPS agar added with 2% of glucose and inoculated with a 0.5 MacFarland standard from identified strains. Agar plates were incubated at 32 °C for a variable time: for *Candida* genus, plates were incubated for 24 h, while for *Cryptococcus* spp., *Rhodotorula* spp., *Saprochaete* spp. and *Saccharomyces* spp. incubation was prolonged up to 48 h. ATCC control strains such as *C. krusei* (6528) and *C. parapsilosis* (22019) were also tested. MIC was evaluated as the intersection point between the elliptic inhibition halo and the highest concentration along the antifungal strip. MIC values from the MIC strip test were rounded up to the next two dilutions, to obtain values more confrontable with broth dilution ones. The essential agreement between the two methods was calculated within + one- or two-fold dilution. Statistical analyses for geometric mean, mode and essential agreement were performed using the MedCalc Statistical Software version 17.9.2 (MedCalc Software bvba, Ostend, Belgium, 2017; http://www.medcalc.org; accessed on 21 December 2022).

## Figures and Tables

**Figure 1 antibiotics-12-00251-f001:**
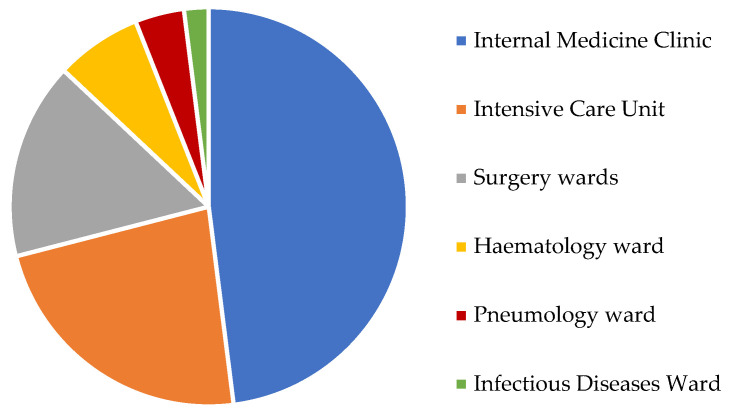
Isolates distribution among hospital wards.

**Figure 2 antibiotics-12-00251-f002:**
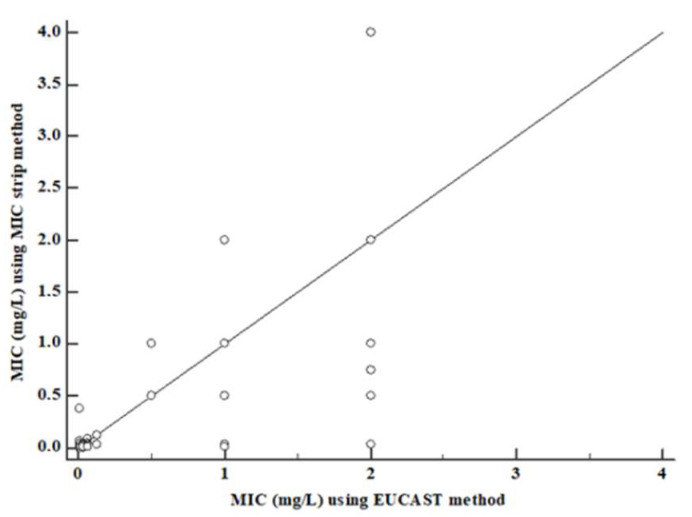
Essential agreement between the EUCAST reference method and MIC strips.

**Table 1 antibiotics-12-00251-t001:** Isavuconazole in vitro susceptibility data collected for enrolled yeast clinical isolates both through EUCAST broth and MIC Strip methods.

	N (%)	EUCAST (E. Def. 9.3)					MIC Strip Test				
		MIC Range (mg/L)	MIC_50_	MIC_90_	ECOFF (mg/L)	% >ECOFF	MIC Range (mg/L)	MIC_50_	MIC_90_	ECOFF (mg/L)	% >ECOFF
*C. albicans*	283 (44.9%)	0.008–1	0.008	0.016	0.125	0.70	0.004–0.125	0.004	0.032	0.125	0
*C. glabrata*	53 (8.4%)	1–2	2	2	-	-	0.5–4	2	2	-	-
*C. parapsilosis* complex	151 (24.0%)	0.008–0.125	0.016	0.016	16	0	0.004–0.032	0.016	0.032	16	0
*C. krusei*	24 (3.8%)	0.5–2	0.5	0.5	16	0	0.5–1	0.5	0.5	16	0
*C. tropicalis*	68 (10.8%)	0.008–2	0.008	0.008	16	0	0.004–0.38	0.004	0.004	16	0
*C. guilliermondii*	12 (1.9%)	0.016–0.032	0.016	0.016	16	0	0.016–0.032	0.008	0.008	16	0
*C. famata*	6 (0.9%)	0.008	0.008	0.008	-	-	0.008	0.008	0.008	-	-
*S. capitata*	12 (1.9%)	2	2	2	16	0	1	1	1	16	0
*S. cerevisiae*	12 (1.9%)	0.03	0.03	0.03	16	0	0.004	0.004	0.004	16	0
*C. neoformans*	5 (0.8%)	0.016	0.016	0.016	16	0	0.008	0.008	0.008	16	0
*Rhodotorula species*	4 (0.6%)	0.016	0.016	0.016	-	-	0.008	0.008	0.008	-	-
*C. krusei* ATCC 6258	-	0.016–0.06	0.03	-	-	-	0.016–0.06	0.03	-	-	-
*C. parapsilosis* ATCC 22019	-	0.008–0.03	0.016	-	-	-	0.008–0.03	0.016	-	-	-

**Table 2 antibiotics-12-00251-t002:** Essential agreement between EUCAST broth method and MIC Strip method for the different yeast species involved in the study.

	N	% Essential Agreement (±2-Fold Dilution)
*C. albicans*	283	99.2%
*C. glabrata*	53	98.1%
*C. parapsilosis* complex	151	100%
*C. krusei*	23	100%
*C. tropicalis*	68	95.6%
*C. guilliermondii*	12	100%
*C. famata*	6	100%
*S. capitata*	12	100%
*S. cerevisiae*	12	100%
*C. neoformans*	5	100%
*Rhodotorula species*	4	100%
Total	629	99.3%

## Data Availability

Not applicable.

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
