# Peer review of "Comparison between EUCAST Broth Microdilution and MIC Strip Test in Defining Isavuconazole In Vitro Susceptibility against Candida and Rare Yeast Clinical Isolates"

_antibiotics, 2023, doi:10.3390/antibiotics12020251_

Round 1

Reviewer 1 Report

I have read with interest the manuscript titled “Comparison between EUCAST broth microdilution and MIC  Strip test in defining isavuconazole in vitro susceptibility against Candida and rare yeast clinical isolates”.

Authors have carried out a similar study using different methods to evaluate in vitro antifungal susceptibility of isovuconazole against Aspergillus.

I have found it is an interesting manuscript, however prior to further processing of the paper several points need to beclarified

-In Results section

- One of the objectives of the study is to analyze the in vitro susceptibility of isavuconazole, which is not yet approved for use in candidiasis, so, in my opinion, it would be interesting to include the sensitivity against conventional treatment, for example fluconazole

-In Material and methods section

- Please separate the methodology into subsections, for ease of reading.

Author Response

  1. Question: In results section, one of the objectives of the study is to analyze the in vitro susceptibility of isavuconazole, which is not yet approved for use in candidiasis, so, in my opinion, it would be interesting to include the sensitivity against conventional treatment, for example, fluconazole.

 Answer: Thank you for the suggestion. We decided to add a few lines within the discussion (lines 312-316), explaining how other azoles in vitro susceptibility through EUCAST broth microdilution could be the target of future investigations on the processed strains.

  1. Question: In Material and methods section please separate the methodology into subsections, for ease of reading.

Answer: Three different subsections have been created along the “Material and methods” paragraph. Thank you for this advice.

Reviewer 2 Report

The manuscript entitled “Comparison between EUCAST broth microdilution and MIC Strip test in defining isavuconazole in vitro susceptibility against Candida and rare yeast clinical isolates” is suitable for Antibiotics Journal. However, some comments will help improve the manuscript.

Introduction:

Line 32-33 “Furthermore, several studies confirm isavuconazole clinical
efficacy in case of invasive aspergillosis, mucormycosis and candidaemia.” Reference missing? Perhaps the references to the previous sentence (5-7) should be placed here.

Line 33-35. “So” is too informal, I suggest changing this connector or rewriting the sentence. Also, this statement should be referenced.

Line 42. Remove the space after the parentheses.

Results:

Regarding the results, normally the figures and tables should be placed just after being referenced or in the same section and not all at the end.

On the other hand, although I understand that they present the data and are then summarized in a fairly complete table (Table 1). Sections 2.2 and 2.3 are too long and difficult to read, presenting the same information and data as in Table 1. I suggest developing the information in these sections a bit to make it more attractive, even being able to compare the different data obtained from the isolated ones or at least separate the information in these sections to make it more readable.

Table 1: “C.famata” need a space. Could there be a small gap in the underlined bar that exists between EUCAST and MIC strip test? I think this change would improve the display of the table.

Discussion:

The discussion is well written and developed.

Line 260. C. krusei. “.” Missing.

Material and methods:

Regarding to Material and Methods, I suggest that section 4 of MMs be divided into subsections that separate the biological material used from the different experiments carried out.

In addition, they comment that all the species were previously identified before their use (Line 304). Could you comment on how they were identified?

Line 331. ATCC should be added to the control strains as before.

References:

Is the reference 33 missing?

Author Response

  1. Question: Line 32-33 “Furthermore, several studies confirm isavuconazole clinical
    efficacy in case of invasive aspergillosis, mucormycosis and candidaemia.” Reference missing? Perhaps the references to the previous sentence (5-7) should be placed here.

Answer: The 5-7 references have been placed after the sentence “Furthermore, several studies confirm isavuconazole clinical efficacy in case of invasive aspergillosis, mucormycosis and candidaemia.” Thank you for the observation.

  1. Question: Line 33-35. “So” is too informal, I suggest changing this connector or rewriting the sentence. Also, this statement should be referenced.

Answer: Thank you for your observation. The term “so” has been replaced with the more formal word “therefore”.

  1. Question: Line 42. Remove the space after the parentheses.

Answer: We apologize for this typo. The space has been removed.

  1. Question: Regarding the results, normally the figures and tables should be placed just after being referenced or in the same section and not all at the end.

Answer: Figures and tables have been correctly placed. We would like to apologize for the mistake.

  1. Question: On the other hand, although I understand that they present the data and are then summarized in a complete table (Table 1). Sections 2.2 and 2.3 are too long and difficult to read, presenting the same information and data as in Table 1. I suggest developing the information in these sections a bit to make it more attractive, even being able to compare the different data obtained from the isolated ones or at least separate the information in these sections to make it more readable.

 Answer: We decided to not remove any information from the results, considering all the data essential to correctly evaluate the manuscript. However, several spaces have been added to make 2.2 and 2.3 sections more readable, according to the reviewer’s kind suggestion. 

6. Question: Table 1: “famata” need a space. Could there be a small gap in the underlined bar that exists between EUCAST and MIC strip test? I think this change would improve the display of the table.

 Answer: The space in “C.famata” has been fixed. A small gap in the underlined bar that exists between EUCAST and MIC strip test has also been added.

  1. Question: Line 260. C. krusei. “.” Missing.

 Answer: It has been added. We would like to highlight that the expression is now placed in line 283.

  1. Question: Regarding to Material and Methods, I suggest that section 4 of MMs be divided into subsections that separate the biological material used from the different experiments carried out.

Answer: Three different subsections have been created along the “Material and methods” paragraph. Thank you for this advice.

  1. Question: In addition, they comment that all the species were previously identified before their use (Line 304). Could you comment on how they were identified?

Answer: The sentence “Species or genus identification were confirmed through MALDI Biotyper® Sirius System (Bruker) for all the selected isolates” has been added (lines 331-332).

  1. Question: Line 331. ATCC should be added to the control strains as before.

 Answer: It has been fixed. Thank you for the observation (line 346 and line 362).

  1. Question: Is the reference 33 missing?

Answer:  We would like to apologize for this lapse. Reference 33 has been correctly placed along the discussion.

Round 2

Reviewer 1 Report

After reading the last version of manuscript “Comparison between EUCAST broth microdilution and MIC  Strip test in defining isavuconazole in vitro susceptibility  against Candida and rare yeast clinical isolates” I have found that the authors made the suitable changes, so the article has improved substantially and can accept in current form.